# Effects of Temperature and Glyphosate on Fatty Acid Composition, Antioxidant Capacity, and Lipid Peroxidation in the Gastropod *Lymneae* sp.

**Mariem Fadhlaoui * and Isabelle Lavoie**

Centre Eau Terre Environnement, Institut National de la Recherche Scientifique, 490 Rue de la Couronne, Québec, QC G1K9A9, Canada; isabelle.lavoie@ete.inrs.ca
* Correspondence: mariem.fadhlaoui@ete.inrs.ca

**Abstract:** Little is known about the potential effects of glyphosate on freshwater gastropods and possible interactions between glyphosate and other stressors. A two-way factorial experiment was conducted to investigate the effects of temperature (20 °C/25 °C) and glyphosate (0 µg/L/200 µg/L) on *Lymnaea* sp. After 21 days, antioxidant capacity (superoxide dismutase (SOD), catalase (CAT), glutathione peroxidase (GPx) and glutathione-S-transferase (GST)), malondialdehyde content (MDA), and fatty acid (FA) composition of *Lymnaea* sp. tissue were measured. Temperature had an effect on SOD activity and GPx activity. In contrast, an increase in GST activity was observed in glyphosate-exposed snails, highlighting the role of GST in the glyphosate detoxification process. Differences in temperature and glyphosate did not affect lipid peroxidation (MDA); however, we observed a trend suggesting the presence of higher MDA content in glyphosate-exposed snails at 20 °C. The FA groups were generally not strongly affected by the treatments, except for omega−9 (n-9) that was markedly lower at the higher temperature. Changes were also observed in individual FA as a response to glyphosate and/or temperature. For example, a significant decrease in 18:1n9 was observed at 25 °C. Our results showed that antioxidant capacity and FA profiles were mainly affected by temperature, while glyphosate seemed to have a lesser impact.

**Keywords:** snails; glyphosate; antioxidant enzymes; lipid peroxidation; fatty acids; temperature

## 1. Introduction

Freshwater ecosystems exposed to agricultural, industrial, and urban land uses can have increased concentrations of contaminants, particularly pesticides. Glyphosate is one of the most commonly used pesticides and is frequently detected in surface waters around the globe [1]. Glyphosate is a non-selective herbicide often used in agriculture, horticulture, and garden maintenance and can enter surface waters through overland runoff. Glyphosate is also found in cotton fiber and has been detected in effluents from the textile industry [2,3]. Because of widespread use, relatively high concentrations of glyphosate have been detected in various aquatic ecosystems. For example, glyphosate concentrations reached 40.8 µg/L in surface waters in Canada [4], 700 µg/L in Argentina [5], and 430 µg/L in the USA [6].

Glyphosate is highly soluble in water and thus is readily taken up by aquatic organisms [7]. Several studies have demonstrated that glyphosate exhibits a wide range of effects on aquatic organisms. In fish, glyphosate can induce histological alteration in the gills, kidneys, and liver [8], inhibit enzymes and alter metabolism [9,10], and cause genotoxic alterations such as nuclear abnormalities and DNA degradation [11]. In amphibians, several studies have shown that glyphosate can cause increased mortality, decreased growth and body size, metamorphosis perturbations, and gonadal abnormalities [12–14]. In molluscs, glyphosate can affect energy metabolism, and alter growth rates, condition indices, sexual maturity, and enzymatic activities [15–17]. In crustaceans, glyphosate exposure has been shown to induce oxidative stress and DNA damage, inhibit antioxidant and immune

responses, and affect metabolic efficiency [18–20]. Lastly, for algae, studies of glyphosate exposure on several different species have reported toxicity in regard to growth rates, photosynthesis, and chlorophyll-a synthesis [21,22].

Similar to other contaminants (e.g., metals), glyphosate is also known to induce excessive reactive oxygen species (ROS) such as hydroxyl radical ($\cdot$OH), superoxide anion ($O_2^{\cdot-}$), singlet oxygen ($O_2^{\cdot}$), and hydrogen peroxide ($H_2O_2$) [23]. The equilibrium between ROS production and elimination is crucial for organism homeostasis. An imbalance between pro-oxidants and antioxidants leads to oxidative stress and excessive ROS, resulting in irreversible macromolecule (i.e., lipids, proteins, and nucleic acids) damage. Under oxidative stress, different antioxidant enzymes are involved in the elimination of ROS. Superoxide dismutase (SOD) and catalase (CAT) are the first line of antioxidant defense. Similarly, glutathione peroxidase (GPx) can metabolize many different peroxide species and glutathione S transferase (GST) catalyzes the conjugation of a variety of metabolites [24]. The degradation of cellular macromolecules, mainly lipids [25], by ROS is considered the most important cause of cell dysfunction and can further result in the formation of aldehydes (i.e., malondialdehyde) from polyunsaturated fatty acids (PUFA) [26]. The effects of glyphosate on these cellular compounds and enzymes have been studied in different aquatic organisms, particularly fish [27,28] and molluscs [29,30]. Although responses differed depending on the species and/or tissues sampled, past studies suggest that the use of these cellular compounds and enzymes as biological endpoints can provide useful information on toxicological effects of glyphosate on aquatic biota.

Another biological descriptor that is sensitive to a variety of environmental stressors is organism or tissue fatty acid composition. Fatty acids (FA) play a key role in aquatic food webs as one of the most important resources transferred across trophic levels [31]. Fatty acids are involved in many different biochemical processes, are a source of energy, and are the main constituent of cellular membranes [32,33]. Some FAs, including long chain polyunsaturated FA (LC-PUFA) such as eicosapentaenoic (EPA, 20:5n3), arachidonic acid (ARA, 20:4n3), and docosahexaenoic (DHA, 22:6n3) are only synthesized by primary producers. Consumers must obtain these essential FA through trophic interactions because these molecules cannot be synthesized in a sufficient quantity [34]. Many contaminants are also known to induce excessive ROS production, which can initiate the peroxidation and degradation of lipids (LPO) [23,35]. Lipids with greater proportions of unsaturated FAs are more prone to LPO than FAs with a low number of double bonds. For example DHA, with six double bonds, is nearly 7.5 fold more prone to peroxidation than a FA with two double bonds [36]. The alteration of FA profiles after exposure to organic and inorganic contaminants has been investigated in some aquatic organisms. In brown trout (*Salmo trutta*) the amount of omega$-$3 LC-PUFAs, EPA, and DHA decreased after exposure to glyphosate (10 and 20 mg/L over a 30 day experiment [37]. Similarly, in marine diatoms (*Phaeodactylum tricornutu*) FA profiles were altered after glyphosate exposure (10, 50, 100, and 250 µg/L), as indicated by a decrease in the EPA and omega$-$6/omega$-$3 ratio [38].

In addition to contaminant exposure, aquatic ecosystems are also subject to increasing water temperature as a result of landscape alteration and global climate change (e.g., warming climate, increased impervious surfaces) [39]. Warmer waters may affect various metabolic functions that stimulate ROS production and have the potential to alter FA composition. For example, FA composition in poikilothermic organisms can change alongside temperature in order to maintain membrane functions and properties. According to homeoviscous adaptation theory [40], greater temperatures will result in a decrease in the proportion of unsaturated FA in cell membranes. The wide range of studies showing FA profile changes under various anthropogenic stresses suggests that the use of this biological descriptor could be a valuable biomarker in assessing the effects of environmental perturbations on aquatic organisms and food webs.

Ecotoxicological studies have often been performed to evaluate the effects of individual stressors on aquatic species. However, because of widespread anthropogenic change, it is increasingly important to consider the effects of multiple stressors. For example,

warmer water temperatures may increase the sensitivity of organisms to contaminants and exacerbate the effects of ROS overproduction (i.e., lipid peroxidation, cellular damage, altered metabolic functions) from pesticide exposure. The response of various organisms to thermal stress and glyphosate exposure has been examined independently [29,41] but few studies have investigated their combined effects, particularly using several biological descriptors. The aim of our study was to determine the combined effects of glyphosate and temperature on the freshwater snail *Lymnaea* sp. Our experiment used a 2 by 2 factorial approach where snails were exposed to glyphosate and increased water temperatures. Effects were evaluated based on multiple physiological end-points: (1) antioxidant enzymes activities (SOD, CAT, GST and GPx), (2) lipid peroxidation (MDA), and (3) FA composition.

## 2. Materials and Methods

### 2.1. Experimental Design

Experimental units consisted of 1.5 L plastic microcosms (n = 12) filled with 1 L dechlorinated tap water which were equipped with aeration pumps and covered with plastic lids. Prior to the start of the experiment, snails (*Lymnaea* sp.) of similar sizes (13.1 ± 1.5 mm shell length) were first placed in a larger microcosm for 7 days to acclimate them to laboratory conditions. After the acclimation period, 20 snails were randomly selected and placed into each experimental microcosm. Four experimental treatments (n = 3) were then established: no glyphosate at 20 °C (Ctl−20 °C), no glyphosate at 25 °C (Ctl−25 °C), glyphosate at 20 °C (Gly−20 °C) and glyphosate at 25 °C (Gly−25 °C) (Figure 1). Water temperature was maintained at 20 °C (room temperature) for the two lower temperature treatments and increased over a 5-day (1 °C per day) period then maintained at 25 °C for the two higher temperature treatments. Within each temperature treatment, half of the microcosms were uncontaminated (control) and half were contaminated with glyphosate to a final concentration of 200 µg/L (chosen to represent elevated environmentally realistic concentrations). To contaminate the microcosms, a glyphosate stock solution was prepared in ultra-pure water using the pure molecule (N-(Phosphonomethyl) glycine; $(HO)_2P(O)CH_2NHCH_2CO_2H$; molecular weight 169.07; Sigma-Aldrich, Canada) and was added to the respective microcosms. All solutions were prepared in plastic material to avoid glyphosate adsorption. During the experimental period (21 days), snails were fed daily with commercial flakes and water (50%) was renewed every two days. Temperature and pH were measured daily to maintain constant values. At the end of the experimental period, snail soft tissues were separated from the shell, immediately frozen in liquid nitrogen, and stored at −80 °C.

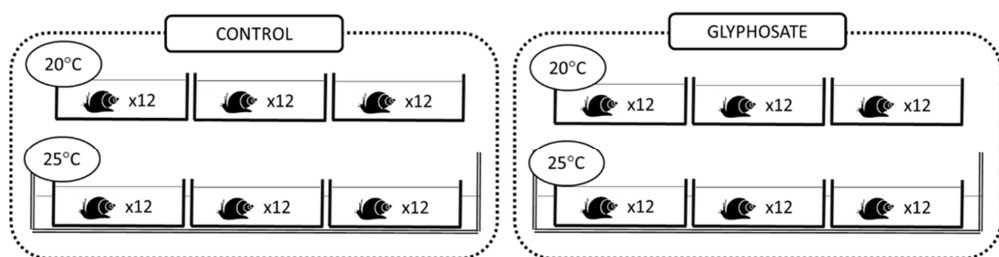

**Figure 1.** Experimental design with microcosms under control and contaminated conditions at two temperatures. Each microcosm contained 12 snails.

### 2.2. Lipid Extraction and Fatty Acid Analysis

Total lipids were extracted from the soft tissue of the entire organism according to Folch et al. (1957) [42]. Two snails per microcosms were used for FA analyses. The detailed procedure for lipid extraction is described in Fadhlaoui et al. (2020) [43], except for the sonication step which was not necessary for snail soft tissue samples. Briefly, total lipids were extracted in a chloroform/methanol mixture (1V/2V) and esterified in boron trifluoride (BF3, 4% methanol) to obtain fatty acid methyl esters (FAME). The resultant FAMEs were analyzed by gas chromatography-flame ionization detector (GC-FID) and

the relative FAME content was determined by comparing chromatograms with reference standards (mixtures of 37 fatty acids, NHI-F, fatty acid methyl ester mix, PUFA NO. 2, animal source, and fatty acid methyl esters kit (Sigma-Aldrich, Oakville, ON, Canada)).

*2.3. Enzyme Assays*

Soft tissue samples were thawed on ice and homogenized in 10 volumes of ice-cold buffer (pH 7.5; 20 mM HEPES; 1 mM EDTA; 0.1% Triton X-100) using an Ultra Turrax T25 homogenizer (Janke and Kunkel, IKA-labortechnik, Staufen, Germany). Immediately after homogenization, GST was analyzed and three aliquots were subsequently stored at –80 °C for SOD, CAT and GPx analyses. Enzymes activities (SOD, CAT, GPx, and GST) were analyzed from homogenized aliquots using a 96-well microplate UV–vis spectrophotometer (Varian Cary 100; Varian Inc., Palo Alto, CA, USA). The activities of SOD (EC: 1.15.1.1), CAT (EC 1.11.1.6) and GPx (EC 1.11.1.9) were analyzed using assay kits 706002, 707002, and 703102, respectively (Cayman Chemical Company Inc, Ann Arbor, Michigan USA). Aliquots for total SOD activity were centrifuged at 4 °C at $5000\times$ g for 5 min and analyzed by the quantification of superoxide radicals generated by xanthine oxidase and hypoxanthine. For the quantification of CAT activity, homogenized aliquots were centrifuged for 15 min at 10,000 g and activities were determined using a colorimetric assay by measuring formaldehyde formed with 4-amino-3-hydrazino-5-mercapto-1,2,4-triazole as chromogen. Lastly, an aliquot for GPx activity was centrifuged at 10,000 g for 15 min. GPx activity was determined from the measure of $NADP^+$, which is directly proportional to GPx. GST activity (EC 2.5.1.18) was measured by following the formation of S-2,4-dinitro phenyl glutathione conjugate at 340 nm according to Habig et al. (1974) [44]. Final activities were expressed in UI (µmol of converted substrate/min)/mg protein in soft tissue.

*2.4. Protein Content*

Enzyme activity is expressed based on protein content. Protein concentration was thus measured on the same aliquot as for each enzyme according to Bradford (1976) using bovine serum albumin as a standard (Thermo Scientific, 23209, Ottawa, ON, Canada). Absorbance was measured at 590 nm and protein concentration was expressed in mg protein/g wet weight.

*2.5. Lipid Peroxidation*

Peroxidative damages to lipids were determined by measuring thiobarbituric acid reactive substances (TBARS) according to the instructions provided with the assay kit (10009055) purchased from Cayman Chemical Company Inc. (Ann Arbor, MI, USA) in supernatant fraction after a centrifugation (1600 g for 10 min at 4 °C) in radioimmunoprecipitation assay buffer (RIPA buffer). Malondialdehyde (MDA) forms complex (MDA-TBA) and was measured by colorimetry at 530 nm. The final concentration was expressed in nmol/mg protein.

*2.6. Statistical Analysis*

A two-way analysis of variance (ANOVA) was used to assess the effects of temperature and glyphosate on all biological descriptors (FA profiles, antioxidant enzymes (SOD, CAT, GST, and GPx), and MDA concentration) as well as to assess interaction between the two factors. Data were expressed as mean $\pm$ standard error of the mean (SEM) (n = 6; 2 snails per triplicated condition) and statistical analyses were conducted using JMP (SAS Institute Inc., Cary, NC, USA). Data distributions were visually inspected and were found to meet the ANOVA assumptions of normality and equal variances.

**3. Results and Discussion**

*3.1. Effects of Temperature and Glyphosate on Snail Enzyme Activity and Lipid Peroxidation*

Two-way ANOVAs performed on enzyme activities (GPx, SOD, GST, CAT) and lipid peroxidation (MDA) revealed no significant interactions between temperature and

glyphosate (Figure 2), but the treatment interaction for GPx was indicative of a trend and was significant at an increased probability of type I error ($\alpha = 0.1$). Our results showed that the fixed effect of temperature significantly increased mean GPx and decreased mean SOD, whereas GST significantly increased under the glyphosate treatment. Glyphosate and temperature did not significantly affect CAT and MDA (Table 1).

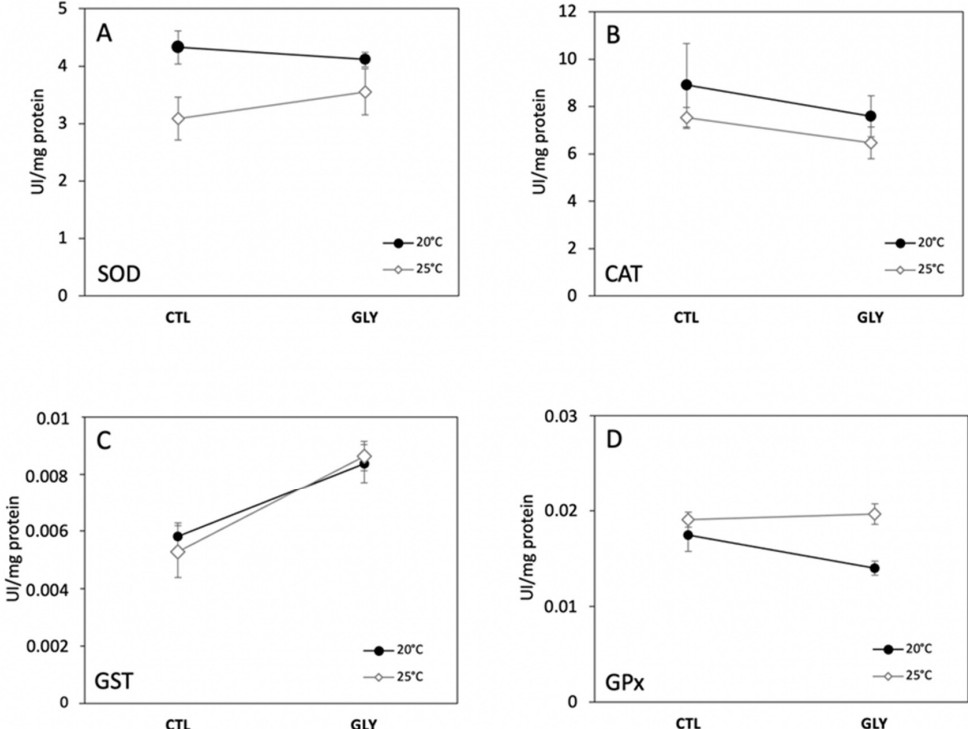

**Figure 2.** Antioxidant enzymes activities in snails (*Lymneae* sp.) (SOD (**A**), CAT (**B**), GST (**C**) and GPx (**D**)) among different exposure conditions at 20 and 25 °C (CTL: control group and GLY: glyphosate exposed group). Values are expressed as mean ± SEM (n = 6 for GST and GPx and n = 5 for SOD and CAT).

**Table 1.** Results from the fixed effect two-way analyses of variance (ANOVAs) with * $p < 0.1$; ** $p < 0.05$; *** $p < 0.005$. Glyphosate (GLY), temperature (TEMP), glyphosate*temperature interaction (GLY*TEMP).

|  | GLY | TEMP | GLY*TEMP |
|---|---|---|---|
| GPx |  | *** | * |
| SOD |  | ** |  |
| GST | *** |  |  |
| CAT |  |  |  |
| MDA |  |  |  |

### 3.1.1. Temperature Effect on Antioxidant Enzymes and Lipid Peroxidation

We found that increased water temperature differentially affected the antioxidant enzymatic activities of freshwater snails (Figure 2). Relative to the control group maintained at 20 °C (Ctl−20 °C), the control group at 25 °C (Ctl−25) showed a decreasing trend in SOD (Figure 2A), while GPx showed the reverse trend (Figure 2D). CAT (Figure 2B) and GST activities (Figure 2C) also declined with increased temperature, but were not significantly different between treatments. Increased temperature can stimulate metabolic rates and induce the release of ROS through the electron transport chain affecting different antioxidant enzymes [35]. According to previous reports [45,46], increased temperature can also accelerate metabolic processes associated with detoxification and xenobiotics

excretion, yet we observed a general inhibition of enzymatic activities for SOD. Monari et al. (2007) [47] investigated the effect of temperature (20 °C, 25 °C, and 30 °C) on the clam *Chamelea gallina* and also observed that SOD activity was lowest at warmer temperatures. In contrast, Verlecar et al. (2007) [48] observed an increase in SOD activity under higher temperatures in the mussel *Perna viridis*. Similarly, Wang et al. (2018) [49] demonstrated that SOD activity gradually increased with temperature in the mussel *Mytilus galloprovincialis*. In our study, decreased SOD activity suggests that this enzyme may be less efficient in eliminating the ROS that were likely over-produced at 25 °C, although we acknowledge that the 5-degree temperature difference between the experimental groups may not necessarily represent thermal-stress.

Our observation of a significant increase in GPx activity in the snails maintained at 25 °C may be related to a post-transcriptional regulation of gpx gene expression. GPx is involved in the detoxification of hydroperoxides, likely produced as a response to thermal stress. The increase in GPx activity suggests an increase in enzyme synthesis as a protective response to ROS production [35]. Similarly, GPx was observed to increase in mussels (*Mytilus galloprovincialis*) maintained at 30 °C after being kept at 15 °C [49]. The increase of GPx activity in snails at 25 °C coincided with a decrease, although not significant, in CAT activity (Figure 2). Overall, the decreasing activity of SOD and GST as a response to temperature suggest that an increase from 20 °C to 25 °C inhibited antioxidant capacity, making organisms potentially more vulnerable by affecting their ability to cope with other environmental stressors.

Increased temperature did not significantly affect lipid peroxidation, although MDA content was slightly higher in control snails kept at 25 °C (Figure 3). Higher temperatures enhance ROS production [35] and consequently increase the risk to lipid peroxidation. The concentration in MDA was used in our study as a biomarker of lipid susceptibility to oxidation. Although non-significant, the observed trend of higher MDA levels in thermally stressed snails (without glyphosate) may be related to an impairment of MDA degradation/elimination or to the failure of antioxidant capacity to inhibit lipid peroxidation [50]. Similar responses were observed in different aquatic invertebrates under thermal stress. For example, Taylor et al. (2017) [51] showed an increase in MDA content at higher temperatures in the bivalve *Anadara trapezia* from 10 to 30 °C. Similarly, Madeira et al. (2014) [52] showed that lipid peroxidation was sensitive to thermal stress and increased with water temperature in three crab species. Thus, the observed increase trend in MDA (although not significant) in the Ctl−25 °C condition could be related to an increase in ROS production that was not counterbalanced by antioxidant enzymes such as SOD and CAT. SOD and CAT are considered the first line of defense to ROS [53] and were observed to decrease with thermal stress. The induction of GPx activity suggests a response to increased lipid peroxidation at 25 °C, and may have been stimulated by higher levels of lipid peroxides. However, the fact that MDA concentrations were higher at 25 °C for the snails that were not exposed to glyphosate (non-significant trend) suggests that GPx activation may not have been sufficient to cope with lipid peroxidation.

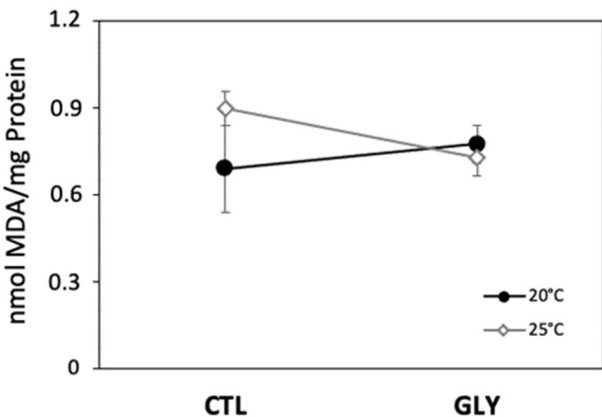

**Figure 3.** Lipid peroxidation expressed as the lipid peroxidation (MDA) concentration in snails among different exposure conditions at 20 °C and 25 °C (CTL: control group and GLY: glyphosate exposed group). Values are expressed as mean ± SEM (n = 6).

### 3.1.2. Glyphosate Effect on Antioxidant Enzymes and Lipid Peroxidation

We found that glyphosate contamination significantly increased GST activity in snails under both temperature conditions (Figure 3; Table 1). Previous studies have demonstrated that glyphosate can interrupt mitochondrial phosphorylation, increase membrane permeability, and trigger ROS accumulation [54,55]. However, most studies on fish and few have investigated the effects of glyphosate on gastropods. Thus, our observation that glyphosate exposure altered GST activity suggests an increase in ROS in contaminated snails. GST is crucial in detoxification and conjugation with glutathione increases the solubility of several endogenous and exogenous toxic compounds, making them easier for organisms to eliminate [56]. Several studies have also shown an increase in GST activity under exposure to different contaminants (e.g., Farcy et al. 2011 [57]; Damiens et al. 2004 [58]). However, glyphosate is not a lipophilic compound and, therefore, its elimination is also possible without biosynthetic conjugation with glutathione. In the present study, GPx, SOD, CAT, and MDA did not vary significantly under the effect of glyphosate. However, we observed a non-significant decreasing trend in SOD activity in glyphosate exposed-snails at 20 °C, and a reversed trend at 25 °C where SOD increased compared to non-contaminated snails. This increase observed at 25 °C is consistent with dos Santos et al. (2014) [59], who reported an increase in SOD activity in different organs of the Asian clam *Corbicula fluminea* exposed to different glyphosate concentrations. ROS overproduction leads to an increase in SOD and results in a simultaneous increase in $H_2O_2$ concentration triggering the activity of CAT, GST, and GPx to neutralize these ROS. However, in our study, CAT activity showed a decreasing trend (although not a significant one) at both temperatures. Similarly, Barky et al. (2012) [60] showed a reduction in CAT in *Biomphalaria alexandrina* snails exposed to atrazine and glyphosate for four weeks. Because we did not see strong effects of GPx, SOD, CAT, and MDA, we suggest that GST has the most important role in glyphosate detoxification.

### 3.1.3. Interaction Effects

GPx levels suggest different responses to glyphosate exposure between the two temperature treatments (20 °C and 25 °C) based on an increased probably of type I error in interaction effects of temperature and glyphosate (Table 1). We also observed interesting trends in temperature*glyphosate interactions, although they were not significant. We therefore recommend that the following discussion should thus be interpreted with caution. At 20 °C, higher MDA concentrations in glyphosate-exposed snails coincided with lower GPx, despite the significant increase in GST activity. We hypothesize that the decrease in antioxidant enzyme activity resulted in the accumulation of free ROS (e.g., superoxide anion and hydrogen peroxide) and increased hydroxyl radical (·OH)

production via Fenton and Haber–Weiss reactions. Hydroxyl radicals degrade PUFA in cellular membranes, leading to hydroperoxide formation [24]. In our study, the observed increase in MDA (non-significant trend) reflects an alteration of detoxification machinery in glyphosate-exposed snails impairing their ability to cope with oxidative damage. A similar observation was reveled in *Biomphalaria alexandrina* snails exposed to glyphosate during four weeks, where increased lipid peroxidation was accompanied by a reduction in SOD, CAT, and glutathione reductase [60]. Likewise, dos Santos et al. (2014) [59] revealed an increase in lipid peroxidation in the Asian clam *Corbicula fluminea* following exposure to 10 mg/L of glyphosate. Moreover, in an experiment on freshwater shrimp (*Macrobrachium nipponensis*), Hong et al. (2018) [18] observed an increase in MDA alongside a decrease in SOD and CAT activities after exposure to 2.80 mg/L and 5.60 mg/L of glyphosate. Glyphosate exposure thus seems to have considerable potential to inhibit antioxidant production systems and lead to diverse oxidative damages in freshwater invertebrates.

In contrast, snails exposed to glyphosate at 25 °C had lower MDA concentrations compared to control snails, while GST activities increased with glyphosate exposure and GPx activity was unchanged. Our observation of a decrease trend in MDA suggests that antioxidant enzymes, mainly GST, are not only involved in glyphosate detoxication and $H_2O_2$ elimination, but also in the elimination of lipid peroxides produced under ROS attack. The inverse relationship between MDA and GST activities suggested by the observed trends highlights a potential dual role of GST as detoxicant and antioxidant in glyphosate-exposed snails. A similar response was reported in juvenile oysters (*Crassostrea gigas*) whereby low MDA content was indicative of more efficient glyphosate metabolization by oysters exposed to a gradient of different glyphosate concentrations in a 35 day laboratory experiment [61].

### 3.2. Snails Fatty Acid Composition as a Response to Glyphosate and Temperature

Several studies have investigated various effects of glyphosate exposure on different mollusc species by measuring endpoints related to biomolecules oxidation, changes in antioxidant capacity, and general metabolic enzymes. However, few if any studies have used FA to evaluate pesticides effects. FA composition has been determined in molluscs such as mussels, oysters, clams, scallops, and snails in marine environments [62–64], but little information is available for freshwater gastropods, especially under contaminated conditions.

We identified 35 FA in the total lipid content extracted from snails with the predominant group (51%) being PUFA (Figure 4; Table 2). Linoleic acid (18:2n6), eicosadienoic acid (20:2n6), arachidonic acid (20:4n6), and eicosapentaenoic acid (20:5n3) were the most abundant PUFA representing about 33%. Oleic acid (18:1n9) and eicosenoic acid (20:1n9) represented about 25% of total MUFA, whereas palmitic acid (16:0), arachidic acid (18:0), and lignoceric acid (24:0) accounted for about 30% of total SFA. Slight differences in FA composition were observed between our study and a past study on freshwater snails [65], but are likely related to the origin of the organisms. For example, Fried et al. (1993) [65] collected snails from the underside of decaying leaf litter. Snail FA composition was correlated with the FA composition of their food supply and dominated by C18 PUFA and LC-PUFA [66]. However, in our study, snails were fed with commercial flake food which was reflected in their FA composition as shown in Table A1.

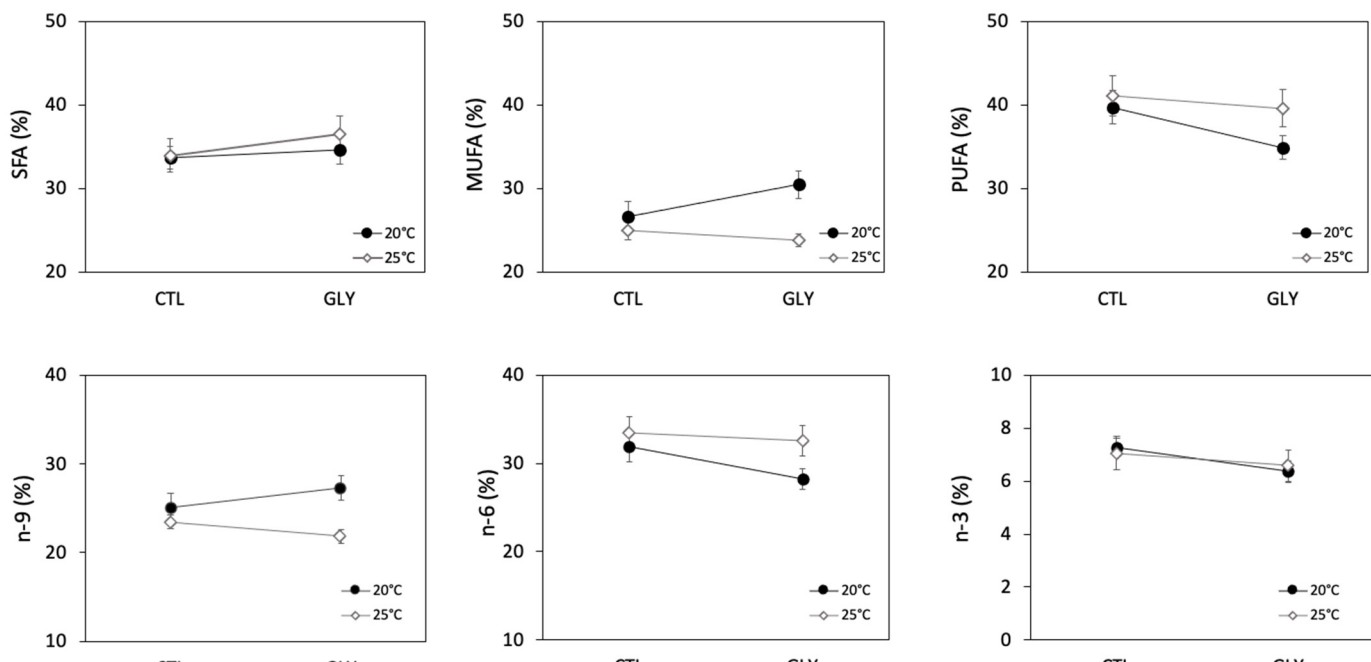

**Figure 4.** Fatty acid groups in snails (*Lymneae* sp.) among different exposure conditions in the control group (CTL) and glyphosate-exposed group (GLY) at 20 °C and 25 °C. Values are shown as % mean ± SEM for each condition (n = 6, two snails per triplicated conditions). Saturated Fatty Acids (SFA), Monounsaturated Fatty Acids (MUFA), Polyunsaturated Fatty Acids (PUFA), omega−9 (n-9), omega-6 (n-6), omega-3 (n-3).

**Table 2.** Fatty acid composition in snails (*Lymneae* sp.) in control group (CTL) and glyphosate-exposed group (GLY) at 20 and 25 °C. Values are shown as % mean ± SEM (n = 6) for each condition. Significant results from the two-way ANOVA are presented with * $p < 0.1$; ** $p < 0.05$; *** $p < 0.005$. Glyphosate (GLY), temperature (TEMP), glyphosate*temperature interaction (GLY*TEMP).

| | 20 °C | | 25 °C | | Two-Way ANOVA | | |
|---|---|---|---|---|---|---|---|
| | **CTL** | **GLY** | **CTL** | **GLY** | **GLY** | **TEMP** | **GLY * TEMP** |
| **14:0** | 0.56 ± 0.05 | 0.50 ± 0.10 | 0.57 ± 0.04 | 0.43 ± 0.05 | | | |
| **15:0** | 0.32 ± 0.03 | 0.26 ± 0.01 | 0.35 ± 0.03 | 0.34 ± 0.02 | | ** | |
| **16:0** | 17.13 ± 0.83 | 18.01 ± 0.61 | 15.60 ± 1.11 | 16.71 ± 0.81 | | * | |
| **17:0** | 0.73 ± 0.07 | 0.66 ± 0.08 | 0.80 ± 0.04 | 0.90 ± 0.04 | | ** | |
| **18:0** | 11.27 ± 0.94 | 10.56 ± 1.05 | 13.37 ± 1.05 | 17.26 ± 0.92 | | *** | |
| **20:0** | 0.49 ± 0.06 | 1.56 ± 0.77 | 0.41 ± 0.04 | 0.34 ± 0.01 | | | |
| **21:0** | 0.10 ± 0.03 | 0.18 ± 0.10 | 0.16 ± 0.01 | 0.09 ± 0.06 | | | |
| **22:0** | 0.53 ± 0.05 | 0.27 ± 0.15 | 1.33 ± 0.05 | 0.19 ± 0.06 | | | |
| **24:0** | 2.54 ± 0.19 | 2.63 ± 0.23 | 1.38 ± 0.23 | 1.84 ± 0.11 | | ** | |
| **16:1n7** | 0.16 ± 0.11 | 0.80 ± 0.18 | 0.16 ± 0.14 | 0.30 ± 0.13 | *** | | |
| **17:1** | 0.95 ± 0.44 | 1.25 ± 0.28 | 0.63 ± 0.40 | 1.40 ± 0.55 | | | |
| **18:1n9** | 22.05 ± 1.59 | 24.28 ± 1.24 | 19.72 ± 1.13 | 18.73 ± 0.88 | | *** | |
| **18:1n7** | 0.40 ± 0.20 | 1.03 ± 0.05 | 0.67 ± 0.18 | 0.24 ± 0.17 | | | ** |
| **20:1n9** | 2.83 ± 0.12 | 2.83 ± 0.18 | 3.41 ± 0.46 | 2.76 ± 0.13 | | | |
| **22:1n9** | 0.16 ± 0.03 | 0.15 ± 0.04 | 0.26 ± 0.03 | 0.15 ± 0.02 | | | |
| **18:2n6** | 12.57 ± 0.86 | 12.94 ± 0.01 | 11.55 ± 0.62 | 9.80 ± 0.59 | | ** | |
| **20:2n6** | 5.98 ± 0.56 | 4.92 ± 0.32 | 6.06 ± 0.53 | 6.53 ± 0.51 | | * | * |
| **22:2n6** | 1.08 ± 0.56 | 2.76 ± 0.89 | 1.62 ± 0.63 | 0.24 ± 0.12 | | | * |
| **18:3n6** | 0.37 ± 0.09 | 0.08 ± 0.08 | 0.39 ± 0.11 | 0.27 ± 0.10 | * | * | |
| **18:3n4** | 0.28 ± 0.07 | 0.22 ± 0.02 | 0.44 ± 0.04 | 0.31 ± 0.07 | | * | |
| **18:3n3** | 0.82 ± 0.02 | 0.69 ± 0.00 | 0.80 ± 0.02 | 0.64 ± 0.03 | ** | | |
| **20:3n6** | 1.91 ± 0.35 | 0.95 ± 0.04 | 1.65 ± 0.12 | 1.65 ± 0.03 | ** | | ** |
| **20:3n3** | 0.67 ± 0.15 | 0.50 ± 0.05 | 0.53 ± 0.18 | 0.33 ± 0.13 | | | |
| **18:4n3** | 0.13 ± 0.02 | 0.11 ± 0.32 | 0.12 ± 0.01 | 0.08 ± 0.00 | * | | |
| **20:4n6** | 10.00 ± 1.52 | 6.60 ± 0.06 | 12.20 ± 1.01 | 13.07 ± 0.04 | | *** | ** |
| **20:4n3** | 0.33 ± 0.07 | 0.23 ± 0.00 | 0.36 ± 0.12 | 0.11 ± 0.01 | ** | | |
| **20:5n3** | 4.90 ± 0.25 | 4.43 ± 0.30 | 4.79 ± 0.32 | 4.53 ± 0.38 | | | |
| **22:5n3** | 0.34 ± 0.15 | 0.35 ± 0.13 | 0.44 ± 0.19 | 0.46 ± 0.22 | | | |
| **Others *** | 0.43 ± 0.05 | 0.22 ± 0.25 | 0.22 ± 0.02 | 0.25 ± 0.15 | | | |
| **U/S Ratio** | 2.00 ± 0.13 | 1.92 ± 0.15 | 2.00 ± 0.18 | 1.65 ± 0.11 | | | |
| **n-9 UFA** | 25.06 ± 1.64 | 27.34 ± 1.38 | 23.48 ± 0.81 | 21.67 ± 0.84 | | *** | |
| **n-6 UFA** | 31.89 ± 1.66 | 28.25 ± 1.12 | 33.47 ± 1.84 | 31.57 ± 1.61 | | * | |
| **n-3 UFA** | 7.28 ± 0.34 | 6.36 ± 0.41 | 7.06 ± 0.64 | 6.18 ± 0.49 | | | |
| **SFA** | 33.66 ± 1.35 | 34.64 ± 1.77 | 33.97 ± 2.03 | 36.56 ± 2.14 | | | |
| **MUFA** | 26.64 ± 1.84 | 30.46 ± 1.64 | 24.99 ± 1.15 | 23.82 ± 0.73 | | *** | * |
| **PUFA** | 39.71 ± 1.99 | 34.90 ± 1.44 | 41.04 ± 2.41 | 39.62 ± 2.27 | | | |

* Others: total of 7 fatty acids detectable (14:1n5, 15:1, 24:1n9, 16:2n4, 16:3n4, 16:4n1, and 22:6n3), none of which represented more than 0.2% of total fatty acids in any treatment).

### 3.2.1. Temperature Effect on Snails Fatty Acid Composition

An increase in temperature induced a significant decrease in MUFA (Figure 4), but no significant differences in SFA and PUFA were observed between treatments (Figure 4, Table 2). Similarly, the 25 °C treatment resulted in a significant decrease in the proportion of the n-9 group, while the proportion of the n-6 group increased and was significant at an increased probability of type I error (alpha = 0.1) (Table 2). Certain individual FA were also observed to respond significantly to the temperature treatments (Table 2). Snails in the 25 °C condition had lower proportions of the SFA 16:0 and 24:0, and higher proportions of 18:0 as compared to the snails in the 20 °C condition. The major MUFA 18:1n9 decreased at 25 °C. Despite the fact that no significant effect of temperature was observed on the PUFA group, temperature seemed to affect certain individual PUFAs (Table 2) where 18:2n6 decreased at 25 °C while 18:3n4 and 18:3n6 increased at 25 °C.

Thermal stress is known to have physiological effects on ectotherm organisms and different cellular responses may occur. One well-known response to an increase in temperature is the remodeling of membrane lipids involving changes in FA composition. Changes in FA composition affect the proportions in SFA and UFA, as well as FA chain length where SFA become more abundant and/or with longer carbon chains at higher temperature to compensate for increased membrane fluidity [67]. For example, in blue mussels (*Mytilus edulis*) and in oysters (*Crassostrea virginica*), increased temperature was found to induce FA remodeling through decreased 22:6n3 and 20:5n3 [68]. Similar FA modifications were observed in two clams species (*Ruditapes decussatus* and *Ruditapes philippinarum*) whereby warming induced a significant decrease in EPA and DHA [69]. In our study, the observed modifications in FA profiles reflect a shift in lipid metabolism. In contrast to many marine organisms, the overall FA composition of lipids (total lipid content per snail) remained constant under thermal stress despite some changes in individual FA. Although we observed significant changes in the proportions of several individual FA, the subtle modifications in FA composition observed in our study suggest that *Lymneae* sp. is not very sensitive to an increase in temperature of 5 °C and that the organism is able to maintain a relatively constant FA composition.

### 3.2.2. Glyphosate Effect on Snails Fatty Acid Composition

No significant effect of glyphosate was observed on the various FA groups (Table 2, Figure 4). However, glyphosate had significant effects on certain individual FA, but these FA were mostly marginal with very low proportions. The different changes in FA profiles in glyphosate-exposed snails suggest an alteration in lipid synthesis. Indeed, it was demonstrated that pesticides alter FA biosynthesis, inhibiting FA desaturation [70] as well as the biosynthesis of long chain FA [71]. At both temperatures, lipid composition in glyphosate-exposed snails changed compared to control snails. Indeed, although the effect of glyphosate was generally not significant on higher level FA groups, our results suggest that unsaturation level decreased, as indicated by the lower U/S ratio as well as the decrease in n-3 and n-6. These trends may be considered as compensatory mechanisms to reduce the susceptibility to lipid peroxidation. Similarly, we observed that eicosadienoic acid (20:2n6) was not affected by glyphosate exposure (although there was a significant interaction at an alpha = 0.1). Unlike the other PUFA, eicosadienoic acid is a long chain FA that can be synthetized de novo in different invertebrate species [72]. Eicosadienoic acid has an unusual structure characterized by isolated double bounds that can replace more sensitive PUFA as a structural FA and may show less sensitivity to oxidative stress damage induced by glyphosate exposure. Overall, our study revealed that certain FA in snails, mostly minor ones, were sensitive to glyphosate, and that this contaminant may have interfered with desaturation-elongation process and inhibited FA synthesis. The observed alteration in FA profiles in exposed snails can be considered as a response to glyphosate-induced stress, as shown in previous studies investigating the effect of organic and inorganic pollutants [73,74].

### 3.2.3. Interaction Effects

Interaction effects of temperature and glyphosate were observed on FAs. For example, MUFA were strongly affected by temperature under the glyphosate treatment (Figure 4). Temperature also seemed to have an influence on the response of individual FAs to glyphosate (and vice-versa). The dominant PUFA 20:4n6 increased at 25 °C, while a marked drop was observed under glyphosate at 20 °C. A similar effect of temperature and interaction was noted for 20:2n6, but at an alpha = 0.1. Glyphosate contamination had an effect on 20:3n6 only at 20 °C.

## 4. Conclusions

Ecotoxicological studies associated with biomarker responses to environmental stressors are important to evaluate perturbated ecosystems. Indeed, no single biomarker can provide a clear evaluation of stressors' effects on ecosystems. The use of a biomarker pool can provide a more useful basis for ecotoxicological interpretation. In the present work, the antioxidant enzymes SOD, CAT, GPx, GST, and MDA (as indicators of lipid peroxidation) were used to evaluate the effects of increased temperature and glyphosate exposure on freshwater snails. Temperature generally showed a stronger effect on these biological endpoints than glyphosate, suggesting that this herbicide does not markedly affect snails antioxidant system (except for GST that showed a strong response). Fatty acid composition also seemed to be mainly affected by temperature, although glyphosate exposure induced significant changes in certain marginal FA. Determining changes in FA profiles in primary consumers may improve assessment of contaminant impacts on aquatic ecosystems. Snails play a key role in aquatic ecosystems as they link primary producers and consumers and, therefore, are valuable biological models to evaluate the effects of environmental stressors on ecosystem health.

**Author Contributions:** Each author made substantial contributions to this paper. M.F. was in charge of experimental conceptualization, laboratory experiments, sample analyses, data processing, and writing. I.L. was involved in the project conception, statistical analyses, reviewing, and editing. I.L. was responsible for funding acquisition and for project administration. All authors have read and agreed to the published version of the manuscript.

**Funding:** Financial support from the Fonds de recherche du Québec–Nature et technologies (FRQNT) and from Natural Sciences and Engineering Research Council of Canada (NSERC) is gratefully acknowledged.

**Institutional Review Board Statement:** Not applicable.

**Informed Consent Statement:** Not applicable.

**Acknowledgments:** We would like to thank Stéphane Moïse from the Laboratoire general at INRS-ETE for glyphosate analysis, and Diane Large and Clément Vignal for their help during the experiment. Special thanks are also due to N. Pearce for editing this manuscript and for providing helpful comments and suggestions. I. Lavoie and M. Fadhlaoui are members of Groupe de Recherche Interuniversitaire en Limnologie (GRIL) and Regroupement des Ecotoxicologues du Québec (EcotoQ), both of which are FRQNT strategic clusters.

**Conflicts of Interest:** The authors declare no conflict of interest.

## Appendix A

**Table A1.** Fatty acid composition of commercial snail diet. Values are shown as %, n = 6.

| Fatty Acid | Mean $\pm$ SEM |
|:---:|:---:|
| 16:0 | 27.88 $\pm$ 0.30 |
| 17:0 | 0.60 $\pm$ 0.03 |
| 18:0 | 1.52 $\pm$ 0.96 |
| 16:1n7 | 0.51 $\pm$ 0.23 |
| 18:1n9 | 35.76 $\pm$ 1.26 |
| 20:1n9 | 1.63 $\pm$ 0.03 |
| 22:1n9 | 1.09 $\pm$ 0.05 |
| 18:2n6 | 21.63 $\pm$ 0.24 |
| 18:3n3 | 0.96 $\pm$ 0.02 |
| 20:5n3 | 3.48 $\pm$ 0.15 |
| 22:6n3 | 2.64 $\pm$ 0.03 |
| Others | 2.37 $\pm$ 0.08 |

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
