# Peer review of "Effects of Temperature and Glyphosate on Fatty Acid Composition, Antioxidant Capacity, and Lipid Peroxidation in the Gastropod Lymneae sp."

_water, doi:10.3390/w13081039_

Round 1

Reviewer 1 Report

Dear authors of the manuscript: Effects of temperature and glyphosate on fatty acid composition, antioxidant capacity, and lipid peroxidation in the gastropod Lymneae sp. It was with pleasure that I read and reviewed the work. The study seems relevant and important, is presented in a very clear form. 
Nowadays, the research on gastropods is crucial and it is known that there is a lack of information regarding this group. It is important to have invertebrate models to use in ecotoxicological studies.
Although I think the outputs selected for the study are relevant, maybe some gene expression analysis should be added

In attach, I send the manuscript with comments for your consideration. 

Author Response

We would like to thank the reviewers for comments and suggestions. We have modified our manuscript accordingly or provided a response when necessary. Gene expression analysis was not possible for this study, but we agree that it would be a good additional marker.

Reviewer 2 Report

The paper deals with the effect of glyphosate at the different temperatures on some parameters of lipid peroxidation and enzymatic antioxidant defence, and fatty acid profile of a gastropod (Lymneae sp.)

The paper is well written and well-structured and contains most of the relevant information about the study results.

The introduction is correct and contains nearly all of the relevant references.

However, some editorial corrections require:

L35-36 It would be better to use the same dimension (μg/L or mg/L) for all of the three citations, even if the original publication added a different one.

L37 highly soluble – in which solvent?

L58-59 GPx and GST are both enzyme families, containing different isoenzymes with different enzyme kinetic parameters and substrate specificity; therefore, plural for would be better

L83 LC-PUFA instead of HUFA would be more correct

L84 & L86 Please use the same format in the same manuscript (μg L-1 or μg/L)

L87 omega6/omega3 ratio

L92 poikilothermic instead of poikilotherms

Materials and Methods chapter requires some modification and a more detailed explanation:

L127 Please explain the method for selection of the glyphosate dose

L132 „water was renewed” – please describe more accurately, e.g., 50% or more. It is a critical point because there is no data about the water's glyphosate content during the trial.

L157-158 Please add the EC number for all three enzymes or none of them. The EC 1.15.1.1. is the EC number of all SOD. Which method was determined in the system, total or only SOD1 or SOD2?

L169 If the enzyme activity was expressed as the soft tissue's protein content, it is not correct because the actual activity was determined in the supernatant fraction of homogenates. Thus, it would be more correct to add the activity per mg of the supernatant fraction's protein content.

L169 UI is an acceptable unit, but its description does not contain that it means conversion per minute

L175-180 Please describe which aliquot (crude homogenate or supernatant fraction after centrifugation) was used to determine MDA.

L181-187 Please describe the statistical analysis more accurately, for instance, which method was used for the homogeneity test and which post-hoc test was used in ANOVA.

Results – this chapter is results and discussion, and it contains some critical points. Some subchapters require re-writing because the explanations are sometimes contradictory and mostly hypothetical without supporting experimental data. In particular, the first part about antioxidant enzyme activities and lipid peroxidation should be re-writing. Explanation about heat-stress is not acceptable due to the low temperature difference between the experimental groups and the relatively long period of the study. In heat-stress conditions, more marked differences can be found.

The second part about the fatty acid profile is more acceptable. However, one critical piece of data did not discuss accurately. The fatty acid composition of the diet reflects the fatty acid composition of the gastropod, but this fact is mentioned very shortly.

Proposals for editorial modifications and re-writing:

L216-217 Heat-induced protein denaturation at 25 oC is not a plausible explanation.

L240 „elevated mitochondrial ROS formation” – this statement did not support the lower SOD activity because, in that case, activation of SOD expression at both gene and protein level would be more possible. The higher MDA content means that in higher temperatures, the antioxidant defence was not adequate to inhibit lipid peroxidation.

L252-253 „GPx activation may not have been sufficient to eliminate H2O2 overproduction” – there are no data about H2O2 formation, and this explanation did not support by the lower SOD activity, which requires for the dismutation of superoxide to hydrogen peroxide.

L265-267 This sentence is correct, but glyphosate is not a lipophilic compound; therefore, its elimination is possible without biosynthetic conjugation with glutathione

L269-270 This reference not relevant for this study because the glyphosate dose was much lower.

L291-292 „inactivation of enzyme activities” – this statement is not acceptable because there are no results about the enzyme protein levels, only activities

In conclusion, the paper requires accurate corrections, and the whole Results chapter should re-write.

Author Response

We appreciate your comments. In attach, you find a document with the different responses. 

Reviewer 3 Report

The authors measure the effects of glyphosate exposure and change in temperature on the water snail Lymnea stagnalis. Shortly summarized the analysed parameters focus on organic pollutant detoxification, ROS defence, and fatty acid composition. The methods applied may me seen as a traditional approach. This traditional approach brings the observed data in line with previous published observations. I like this approach as it gives an idea of the acual impact of glyphosate exposure on this species or on molluscs in general.Ok, people might like to have a more detailed view on specific details. However the strength of the paper is the compareability with other studies. The general impression is that this paper is a piece of thorough investigation. 

Author Response

Thank you for reviewing our manuscript, we appreciate your comments. 

Reviewer 4 Report

The paper written by Fadhlaoui and Lavoie investigate the oxidative stress responses and changes in fatty acid composition in the gastropod  Lymneae sp. induced by exposure to glyphosate  and 2 different temperatures.  Although there are many studies related to glyphosate, its persistence in the environment as well as  the toxicity on humans and animals, the presented paper provides useful and some new  information. In my opinion, the manuscript is well written and the experiments were performed competently. But certain points should be corrected before publication:

  1.  Chemical name of glyphosate should be provided
  2. The results and discussion are written together. Thus, I suggest to add discussion in the title of the section 3.

3 . Tables 1 and 2: It seems that some results are missing

4. The last paragraph in Section 3.1.2. Page 8, row 283:

The statement " that GST has the most important role in defending organisms against H2O2 excess", should be changed. The authors should indicate that GST has the most important role in glyphosate detoxification.

  1. Section 3.2.1. Temperature Effect on Snails Fatty Acid Composition

First sentence is: "An increase in temperature induced a significant decrease in MUFA (Figure 4)...

However, I cannot observe a significant difference in MUFA in CTl-20 and CTl-25 groups from the presented results (2nd graph in Figure 4). Moreover,  the values 26.64±1.84  and 24.99±1.15 are presented for CTl-20 and CTl-25, respectively  (Table 2).

Minor points:

Abstract, row 20: highest should be replaced by higher

Page 2, row 79, "than" is written two times

Page 8, row 293, "radicals" should be deleted

Author Response

We appreciate your comments. You find, in attach, a document with the responses.

Round 2

Reviewer 2 Report

The Authors corrected most of the comments for corrections in the revised version of the manuscript, but some remains open:

L197-202 In the statistical analysis description, the homogeneity test, and post-hoc tests remain missing, requiring an accurate description of ANOVA.

L217-234 In this paragraph, the explanation of results remains hypothetical because there was no heat stress in the present study but did not explain the possible significant differences, for instance, in SOD activity.

L279-280 “suggests potential for competition between CAT and GPx in the degradation of H2O2.” -  This statement is not true because there is no competition between CAT and GPx, but their affinity to H2O2 is different. GPx has a higher affinity to low levels of H2O2 than CAT (see: Baud et al., 2004. J. Neurosci. 24: 1531–1540.)

Author Response

The Authors corrected most of the comments for corrections in the revised version of the manuscript, but some remains open:

L197-202 In the statistical analysis description, the homogeneity test, and post-hoc tests remain missing, requiring an accurate description of ANOVA.

Post-hoc tests were not conducted as explained in our response to the reviewers’ comments. A sentence on ANOVA assumptions was added. (L206)

L217-234 In this paragraph, the explanation of results remains hypothetical because there was no heat stress in the present study but did not explain the possible significant differences, for instance, in SOD activity.

We agree that “heat stress” is not the most appropriate term and changed it for “thermal stress” will be more appropriate (L239)

L279-280 “suggests potential for competition between CAT and GPx in the degradation of H2O2.” -  This statement is not true because there is no competition between CAT and GPx, but their affinity to H2O2 is different. GPx has a higher affinity to low levels of H2O2 than CAT (see: Baud et al., 2004. J. Neurosci. 24: 1531–1540.)

We eliminated this hypothesis (L289)
